# Inherited Metabolic Diseases from Past to Present: A Bibliometric Analysis (1968–2023)

**DOI:** 10.3390/children10071205

**Published:** 2023-07-12

**Authors:** Banu Kadıoğlu Yılmaz, Ayşe Hümeyra Akgül

**Affiliations:** 1Department of Pediatric Nutrition and Metabolism, Faculty of Medicine, Selçuk University, Konya 42250, Turkey; 2Department of Pediatrics, Faculty of Medicine, Necmettin Erbakan University, Konya 42080, Turkey; humeyra.akyurek@gmail.com

**Keywords:** inherited metabolic disease, bibliometric study, newborn screening, gene therapy, metabolomics, molecular genetic analysis, phenylketonuria

## Abstract

Bibliometric studies on inherited metabolic diseases (IMDs) do not exist in the literature. Therefore, our research aims to conduct a bibliometric study to determine the current status, trending topics, and missing points of publications on IMDs. Between 1968 and 2023, we conducted a literature search with the keyword “inherited metabolic disease” in the SCOPUS database. We included research articles in medicine written in English and published in the final section. We created our data pool using VOSviewer, SciMAT, and Rstudio software programs for the bibliometric parameters of the articles that met the inclusion criteria. We performed a bibliometric analysis of the data with the R package “bibliometrix” and BibExcel programs. We included 2702 research articles published on IMDs. The top three countries that have written the most articles in this field are the USA (*n* = 501), the United Kingdom (*n* = 182), and China (*n* = 172). The most preferred keywords by the authors were: newborn screening (*n* = 54), mutation (*n* = 43), phenylketonuria (*n* = 42), children (*n* = 35), genetics (*n* = 34), and maple syrup urine disease (*n* = 32). Trending topics were osteoporosis, computed tomography, bone marrow transplantation in the early years of the study, chronic kidney disease, urea cycle disorders, next-generation sequencing, newborn screening, and familial hypercholesterolemia in the final years of the study. This study provides clinicians with a new perspective, showing that molecular and genetic studies of inherited metabolic diseases will play an essential role in diagnosis and treatment in the future.

## 1. Introduction

Inherited metabolic diseases are rare diseases caused by a genetic mutation and have unique biochemical, clinical, and pathophysiological consequences depending on the specific biochemical pathway affected [1,2]. Clinicians have become more knowledgeable in recent years about the pathophysiology, early diagnosis, and treatment of inherited metabolic diseases because of advances in genetics and biochemistry [3,4,5]. Early diagnosis and treatment’s positive contribution by the recently increasing number of studies gives rise to survival, reduces morbidity in inherited metabolic diseases, and causes the introduction and increase of prenatal diagnosis methods and postnatal screening programs to diagnose these diseases [6,7,8,9]. Sir Archibald Garrod described the first inherited metabolic disease in 1908 as alkaptonuria [10]. He discovered the disease’s metabolic pathway and its relationship with Mendelian inheritance [10]. Thus, the importance of biochemical analysis and genetics in medicine and inherited metabolic diseases was recognized for the first time. Then, in 1934, Ivar Asbjörn Følling discovered phenylketonuria, one of the most prevalent inherited metabolic diseases [11]. In the 1960s, Robert Guthrie found a diagnostic test with his name, and newborn screening started for phenylketonuria [12]. The success of preventing mental retardation with dietary treatment caused newborn screening for phenylketonuria to become widespread worldwide [13]. In the 1990s, with the introduction of tandem mass spectrometry, newborn screening for more than 50 inherited metabolic diseases started to be performed in certain places in the world [14]. In recent years, in some places, lysosomal storage diseases like Krabbe, Pompe, Fabry, Gaucher, Niemann-Pick, and Mucopolysaccharidosis, types I and II, as well as X-linked adrenoleukodystrophy, a peroxisomal disease, were included in the newborn screening program [15]. There is a notable rise in the prevalence of metabolic disorders, identified through advancements in the fields of biochemistry and genetics, and this also causes significant expansions in the scope of the newborn screening program. In the 1960s, 80 inherited metabolic diseases were mentioned, but this number increased to over 1000 in the 2020s [10]. This situation shows the development of inherited metabolic diseases over the years and the importance of following the literature on this subject. Evaluating the literature from a broad perspective regarding the trending topics over the years and the least mentioned topics that perhaps show the need to fill the lack of knowledge in this field is essential.

Bibliometric studies are one of the literature review methods that reveal the current status of publications in a particular research field and evaluate the quantitative and qualitative characteristics of these publications [16]. Bibliometric studies contribute to the development of scientific research by presenting the latest analysis of the publications in a particular research field, the latest current approaches, and the gaps that need to be filled in the literature [17]. Through bibliometric analysis, we can gain insights into various aspects such as trending topics, keywords, authors, journals, countries, etc. [18]. Bibliometric analysis programs such as CiteSpace [19], VOSviewer [20], the R package “bibliometrix” [21], and HistCite [22] are used to visualize bibliometric analyses. The VOSviewer program is frequently used in bibliometric analyses of many clinical studies. This program provides an analysis of the frequency with which two documents are cited together by other documents at the same time (co-citations), two articles referencing a common third article in their bibliographies (bibliographic coupling), the proximity and distance of authors from each other during their joint publication productivity in a field (co-authorship), and the common use of selected keywords and their relationship with each other (co-occurrences of keywords) [20]. Many bibliometric analysis studies have been conducted in different medical fields in recent years [23,24,25,26]. However, our comprehensive literature review revealed a conspicuous absence of bibliometric investigations about inherited metabolic diseases.

Our study aims to conduct a bibliometric analysis of the existing body of published literature on inherited metabolic diseases to shed light on the latest status, current issues, and missing points in this field.

## 2. Materials and Methods

### 2.1. Study Design and Search Strategy

We conducted this study retrospectively and cross-sectionally. Before starting the research, we received ethical approval from the ethics committee of Selcuk University Faculty of Medicine (Date 6 June 2023 Decision No. 2023/292). All authors participating in the study read and signed the Declaration of Helsinki. We carried out the study between 7 June 2023–23 June 2023. We conducted a literature search with the keyword “inherited metabolic disease” in the SCOPUS database between 1 January 1968 and 1 May 2023. We created our data pool using VOSviewer, SciMAT, and Rstudio software programs for the bibliometric parameters of the articles that met the inclusion criteria. We performed authors’ bibliographic coupling analysis, co-occurrences of keywords analysis, co-citation analysis, and co-authorship analysis in the publications included in the study using the VOSviewer software program. While evaluating authors’ bibliographic coupling analysis, we determined that each author should have at least five articles, and their publications should be cited at least five times as criteria. In the co-citation analysis, we accepted the minimum number of citations by the authors as 50. In the analyses made with the VOSviewer program, we showed the bibliometric parameters in the same cluster, which are related to each other, with the same color scale. We performed the bibliometric analysis of the data by creating mapping and table visuals with the VOSviewer, R package “bibliometrix,” and BibExcel programs.

### 2.2. Inclusion and Exclusion Criteria

We included articles in the field of medicine written in English and published in the final section, not in the type of book or book series. We excluded from the study the articles published in non-medicine fields (e.g., Pharmacology, Toxicology, and Pharmaceutics; Nursing; Chemical Engineering; etc.), publications of books and book series, publications of non-research articles (e.g., reviews, case reports, letters to the editor, etc.), articles in the “article in press” stage of which the final part was not published, and the articles published in non-English languages.

### 2.3. Data Collection and Statistics

In this study, we used bibliometric parameters, including the article title, publication year, authors, country of the corresponding author (the first author of the study), the name of the journal in which it was published, keywords, and the number of citations to the article. We categorized the study data we obtained with a computer software program and expressed them numerically without the need for any statistical program.

## 3. Results

### 3.1. Analysis of Data

#### 3.1.1. General Analysis of Published Articles on Inherited Metabolic Diseases

We included 2702 research articles published on inherited metabolic diseases within the specified dates and meeting the inclusion criteria. The flowchart of inclusion and exclusion is shown in detail in Figure 1. In these publications, to which 14,782 authors contributed, the annual growth rate was 3.9%, and the average citations per document were 26.4%. General information about the study is given in Table 1.

#### 3.1.2. Distribution of Main Authors

According to the number of articles on inherited metabolic diseases, Hoffmann GF. (*n* = 30), Kölker S. (*n* = 24), and Wang Y. (*n* = 20) were the authors who wrote the most articles. According to their impacts, Hoffman GF. (H-index = 22), Kölker S. (H-index = 14), and Wajner M. (H-index = 13) were the most influential authors. When the productivity of the authors according to the years is examined, it is seen as high, especially in the last ten years. We evaluated 239 authors with 50 or more citations when co-citations were examined. We found a strong correlation between Hoffmann GF, Kölker S, and Goodman SI, which have a high H-index and a high number of articles. In the bibliographic coupling analysis, Hoffmann GF and Wajner M showed a strong relationship in 2010. We determined that Wang Y and Li X, who have many articles, have intensified their publications in recent years. The number of articles, the influence of the top ten authors who have publications in this field, the distribution of the authors’ article productivity by years, the co-citations analysis of authors, and the bibliographic coupling analysis of authors by years are given in Figure 2.

#### 3.1.3. Distribution of Main Journals

When the number of articles and the impact factor of the journals that publish articles on inherited metabolic diseases are examined, the Journal of Inherited Metabolic Disease (*n* = 169), Molecular Genetics and Metabolism (*n* = 90), and Orphanet Journal of Rare Diseases (*n* = 64) are the first three journals to publish the most articles; the Journal of Inherited Metabolic Disease (H-index = 35), Molecular Genetics and Metabolism (H-index = 30), and Human Molecular Genetics (H-index = 19) were the most influential journals. The number of articles and the impact factor of the top ten journals that publish articles in this field are given in Figure 3.

#### 3.1.4. Analysis of Corresponding Authors’ Countries

When the distribution of the articles on inherited metabolic diseases according to the countries of the corresponding authors is examined, the USA (*n* = 501), the United Kingdom (*n* = 182), and China (*n* = 172) were in the top three places. We found that studies in China, Turkey, and Japan are mainly carried out in a single center, while multicenter studies are mainly carried out in the USA and European countries. Again, when the distribution of countries on the world map according to the article is examined, productivity is high in developed countries such as the USA, European countries, and Australia and low in developing countries such as China. However, the production of articles in this field is low in Russia and the least developed countries in Africa. We found that the USA and the United Kingdom, which produce the most articles, have a strong relationship with other European countries regarding publication productivity. These publications intensified between 2005–2010. We found that China’s publication productivity increased, especially after 2018, but its relationship with other countries in producing collaborative work is still weak. The distribution of the articles according to the countries on the world map, the number of articles in the top ten countries according to the countries of the corresponding authors, the number of single or multi-centered studies, and the document weight of the countries of the co-authors by years and their relationship with each other are given in Figure 4.

#### 3.1.5. Analysis of Keywords

In the evaluation of the keywords preferred by the authors in the research articles on inherited metabolic diseases, according to the number of articles, the most preferred six keywords were newborn screening (*n* = 54), mutation (*n* = 43), phenylketonuria (*n* = 42), children (*n* = 35), genetics (*n* = 34), and maple syrup urine disease (*n* = 32) (Table 2). The most preferred keyword, newborn screening, showed a strong association with mutation and tandem mass spectrometry in publications. We detected the association of the keywords myopathy, mitochondria, and whole-exome sequencing. We also saw the association of the keywords Fabry disease, lysosomal storage disorder, mucopolysaccharidosis, and enzyme replacement therapy as being in a different group. The distribution of the 100 most preferred keywords by the authors and co-occurrences of keywords are given in Figure 5.

#### 3.1.6. Trending Topics

When the trending topics in the field of inherited metabolic diseases are evaluated according to the years, it is seen that bone resorption, computed tomography, and bone marrow transplantation subjects were at the forefront in the first periods of the study. Chronic kidney disease, urea cycle disorders, next-generation sequencing, newborn screening, and familial hypercholesterolemia subjects were preferred in recent years. It is seen that prenatal diagnosis and cardiomyopathy are trending topics that have had a wider distribution over the years. It is observed that studies on the subjects of porphyria, hypercalciuria, ochronosis, and mucopolysaccharidoses have decreased in the last ten years, while studies on the subjects of Fabry disease, gene therapy, lysosomal storage disease, maple syrup urine disease, and phenylketonuria have increased. In addition, physicians working on inherited metabolic diseases have recently focused on diagnostic methods such as newborn screening, next-generation sequencing, and tandem mass spectrometry. The distribution of trending topics by year is shown in Figure 6.

## 4. Discussion

In this bibliometric study, in which we evaluated research articles published on inherited metabolic diseases, we evaluated the latest status of publications on this subject, current issues, and missing points in the literature over a wide period of time. Our study is important as it is the first bibliometric study on inherited metabolic diseases.

Most inherited metabolic diseases show autosomal recessive inheritance [27]. For this reason, it is expected to be more common in least-developed and developing countries with a high level of consanguineous marriage. However, in our study, we found that authors from developed countries such as the USA and the United Kingdom published more articles on inherited metabolic disease. These results were similar to those of many bibliometric studies in other medical fields [28,29,30]. In developed countries, better data recording and automation systems may be a reason for this situation. The novelty and reliability of information are among the publication criteria for journals in scientific research [31]. Robust data recording systems in developed countries may have contributed to the productivity of publications in those countries by ensuring the reliability of the information. In addition, more funds and resources are provided in developed countries for research on the diagnosis and treatment of metabolic diseases, which may be another reason. The fact that the multicenter studies in our study are smaller in countries such as Turkey and China than in the USA and European countries also supports this situation. We found a strong association between the H-index and the joint publication productivity of authors with a large number of articles (e.g., Hoffmann GF, Wajner M). This may be one of the reasons for the high number of multicenter publications in developed countries. In addition, unlike in developed countries, clinicians dealing with inherited metabolic diseases in these countries are gathered in certain centers, and the inadequacy of the country-wide data recording system may cause studies to be conducted in a single center. The inadequacy of studies in developing countries does not mean that inherited metabolic diseases are less common in these countries. It is due to the difficulties in diagnosing it for the reasons described above. In our study, the high rate of joint publications between the USA, the United Kingdom, and European countries can also be explained by the relatively lower population and inherited metabolic disease density in these developed countries. Based on the same idea, it can be thought that multicenter studies are not carried out frequently, considering that there is a sufficient number of cases due to the high number of patients in China, which has a high population density. The increase in the literature, especially in the number of publications from China since 2018, may be an indicator of the increased awareness of inherited metabolic diseases in this country.

In our study, newborn screening was the main topic that clinicians researching inherited metabolic diseases were most interested in. Newborn screening, first initiated in the 1960s to diagnose patients with phenylketonuria, has gradually become a part of countries’ national health policies as an important public health initiative to prevent disability and death [32]. Although the number of diseases investigated varies according to country, the scope of newborn screening has been expanded in recent years with the increase in the number of defined metabolic diseases [10,32]. For example, the issue of including lysosomal storage diseases treated with enzyme replacement therapy in the screening program has come to the fore. However, the presence of patients with borderline enzyme activity, the inability to adequately interpret new gene variants, whether the affected individuals need treatment in the pre-symptomatic stage, and the concerns that the balance between the cost and benefit of treatment cannot be achieved have led to debates about the inclusion of these diseases in newborn screening [10]. In a program organized in New York in 2006, the treatment results of five newborns diagnosed with Krabbe disease after screening 2 million newborns increased the debate on the search for lysosomal storage diseases in newborn screening [33]. There were concerns such as not knowing how many Krabbe patients would become symptomatic, not being able to distinguish between newborns affected and unaffected by the disease, undesired treatment results, treatment complications, and ethical concerns. The interest of clinicians working on inherited metabolic diseases in this field may be their desire to eliminate these uncertainties in newborn screening and bring about an international standard. Early diagnosis can reduce mortality and morbidity by providing early treatment for inherited metabolic diseases. For this reason, screening for the highest number of inherited metabolic diseases in newborns is essential, considering treatment efficacy, ethical issues, and cost-benefit balance. We believe that newborn screening will maintain its popularity in the future.

Phenylketonuria (PKU) was also the starting point for newborn screening as one of the first treated inherited metabolic diseases [32,34]. Patients diagnosed with PKU by newborn screening were treated with a low-phenylalanine diet, preventing neurological damage [34]. However, over time, problems in patients’ adherence to diet and the presence of patients whose neurocognitive functions did not improve despite diet therapy brought up new ideas for developing new treatment strategies [34]. In addition to enzymatic treatments, clinical studies have been carried out in gene therapy in recent years [34,35]. The overprocessing of PKU in the studies published on inherited metabolic diseases may be because it is the first known disease among inherited metabolic diseases, and clinicians have more information about PKU because it is diagnosed more frequently than other diseases because of newborn screening programs. In addition, the recent search for genetic and molecular treatments may have led to increased PKU studies.

Mitochondrial diseases and inherited metabolic diseases with myopathy are diagnosed with whole exome sequencing [36,37]. Therefore, the association of the keywords myopathy, mitochondria, and whole exome sequencing drew attention to our study. The importance of whole exome sequencing in these diseases is that a definitive diagnosis cannot be made with basal metabolic tests. Mitochondrial diseases and metabolic myopathies, among inherited metabolic diseases, are two of the areas where difficulties are experienced both in diagnosis and treatment. In a study of mitochondrial disease-suspected patients screened with whole exome sequencing (WES), 67 out of 113 patients were genetically diagnosed with mitochondrial disease [36]. In this study, as an example of selective screening, the high diagnosis rate draws attention [36]. To diagnose mitochondrial or inherited metabolic diseases and other genetic diseases, many studies have been conducted with WES on newborn screening, screening of patients in intensive care units, or screening in selected groups [36,38,39,40]. In addition, in a study conducted with the method of Rapid-WES, which can give results within 5–14 days, nine patients hospitalized in the intensive care unit were studied, and eight patients were diagnosed with inherited metabolic disease. In the same study, most of these patients were shown to have mitochondrial disease [40]. We think that WES will be used more frequently in diagnosis and screening methods in the future and may even be integrated into newborn screening programs in developed countries. Another co-occurrence of keywords in our study was the combination of Fabry disease, lysosomal storage disease, mucopolysaccharidosis, and enzyme replacement therapy. Enzyme replacement therapy was used for the first time in Gaucher disease in the early 1990s and has now become routine in many diseases such as Fabry disease, Pompe disease, MPS types I, II, IV A, VI, VII, and acid lipase deficiency [41]. Since there is still no better treatment option for these diseases, the publications with these keywords are still numerous and up-to-date.

Inherited metabolic diseases are diseases that occur as a result of genetic disorders. Many different mutations can be seen in the same disease group. In addition, it may show clinical variability according to mutation [42,43,44,45]. Clinicians’ detection of the mutation type plays a role in identifying variants that cause symptomatic disease and may also be a guide in gene therapy [45]. While curable inherited metabolic diseases were diagnosed by newborn screening, many inherited metabolic diseases not included in the screening could not be diagnosed. Over time, with the addition of tandem mass spectrometry to newborn screening, the number of inherited metabolic diseases diagnosed has increased. Examining more genetic variants allows authors to notice different clinical findings in the same disease [46]. Because of this, genotype-phenotype relationships began to be investigated [46]. Detailed diagnostic examinations with whole exome sequencing became routine [46,47]. However, whether some variants of unknown significance caused the disease remained a question mark [46]. Furthermore, emerging metabolomics studies have enabled the discovery of new biomarkers for inherited metabolic diseases [46]. In addition, this metabolomics sheds light on whether mutations are responsible for the disease [46]. We see the journey of the diagnostic approach in inherited metabolic diseases, in which the detection of the accumulated substance is being complemented by enzymatic evaluation, molecular genetic studies, and metabolomics. We think this change parallels the change from dietary treatment and enzyme replacement therapy to gene therapy. The increase in studies on genetics, the use of “next generation sequencing” and “tandem mass spectrometry” in expanded newborn screenings, and the discovery of “metabolomics” explain the fact that these three subjects are current and frequently researched topics in “trending topics” [48]. The most recent treatment approach for metabolic diseases is correcting the primary genetic defect. In this approach, conventional gene therapy is applied by delivering a vector containing the correct coding DNA (cDNA) sequence of the defective gene to the host rather than resolving the endogenous genetic defect [49]. Gene therapy has been successfully applied in preclinical trials of inherited metabolic diseases such as glycogen storage type 1a, familial hypercholesterolemia, ornithine transcarbamylase deficiency, and hereditary tyrosinemia type 1 [45,50,51,52]. With the increasing number of mutation studies in inherited metabolic diseases, the clinical diversity of inherited metabolic diseases may have expanded. At the same time, it may have led to increased genetic studies, such as gene therapy, especially in inherited metabolic diseases that are not treatable. However, in our study, we observed that clinicians’ willingness to research incurable or very rare diseases (peroxisomal disease, Farber disease, cerebrotendinous xanthomatosis, etc.) was weak. The intensive studies of clinicians on treatable inherited metabolic diseases such as glycogen storage disease type 1, methylmalonic acidemia, and Fabry disease have caused complications such as chronic renal failure due to the prolongation of life expectancy to be seen more frequently in these patients and have been over-reported in recent studies. We think that in the future, clinicians will follow the complication issue in inherited metabolic diseases more closely with the advances in molecular diagnostic methods and gene therapy.

### Limitations

Evaluation of publications in a single database is a limitation of our study. Another limitation is searching with a single keyword. For this reason, it is an important limitation that we could not search with the keyword “inborn errors of metabolism,” which is used synonymously with inherited metabolic disease in the literature. Although we created a data set that could evaluate the inherited metabolic diseases in our study, it cannot be said that we have thoroughly analyzed the studies for the subgroups of these diseases. Because the inclusion of an article in our research was directly related to whether the keyword we preferred was also preferred by the author or not, and although our aim in evaluating only research articles in our study was to determine the research trend in inherited metabolic diseases, another limitation is that we did not include case reports and case series publications that refer to individual differences in inherited metabolic diseases. Another limitation of the study is that we did not analyze the impact of publications.

## 5. Conclusions

As a result, the diagnostic process that started with newborn screening has increased the number and clinical diversity of inherited metabolic diseases over the years. Phenylketonuria, the first preventable disease diagnosed with newborn screening and its treatment, has been the subject of greatest interest to clinicians dealing with metabolism. We observed that the diagnosis of inherited metabolic diseases was expanded by newborn screening, then tandem mass spectrometry, next-generation sequencing, and metabolomics. Gene therapy is expected to become more important in the future, complementing conventional therapies such as diet and enzymatic therapy. There is a need for bibliometric studies that include more databases and evaluate more original articles on this subject.

## Figures and Tables

**Figure 1 children-10-01205-f001:**
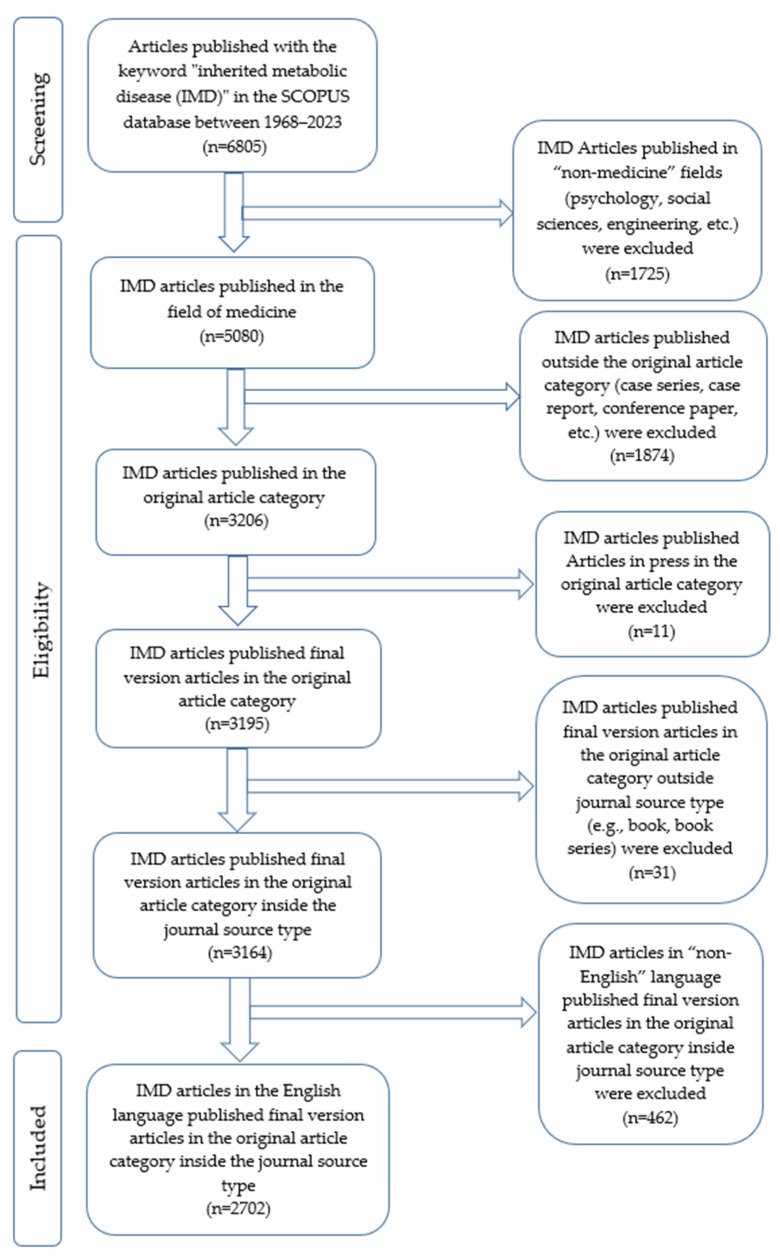
The screening flowchart for articles includes the keyword “inherited metabolic disease”. IMD; inherited metabolic disease; n; number.

**Figure 2 children-10-01205-f002:**
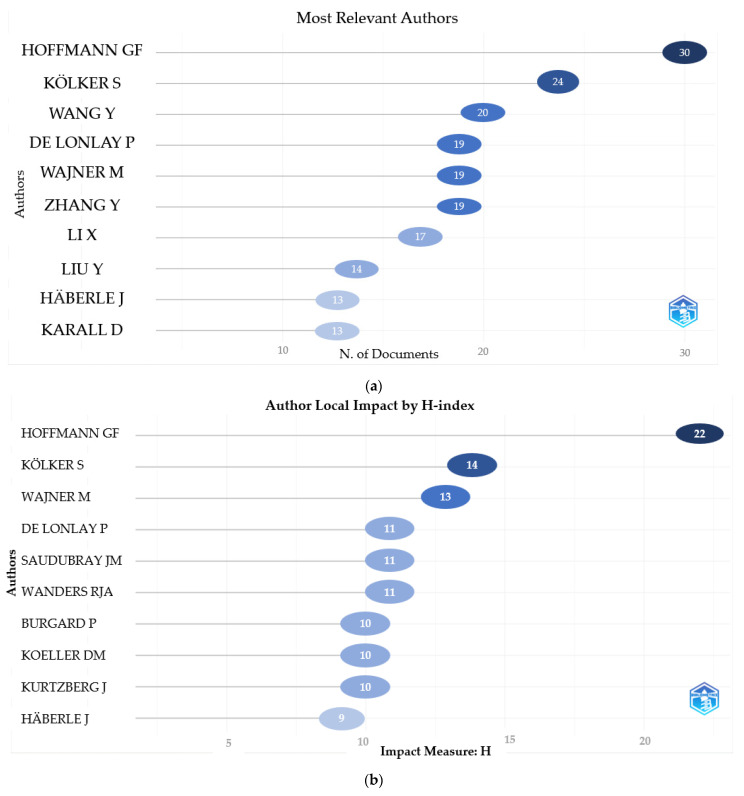
(**a**) The most relevant authors’ number of articles is shown. N; number. (**b**) Author impact by H-index is shown. (**c**) Top-authors production over time is shown. (**d**) A co-citations analysis of authors is shown. (**e**) A bibliographic coupling analysis of authors by years is shown.

**Figure 3 children-10-01205-f003:**
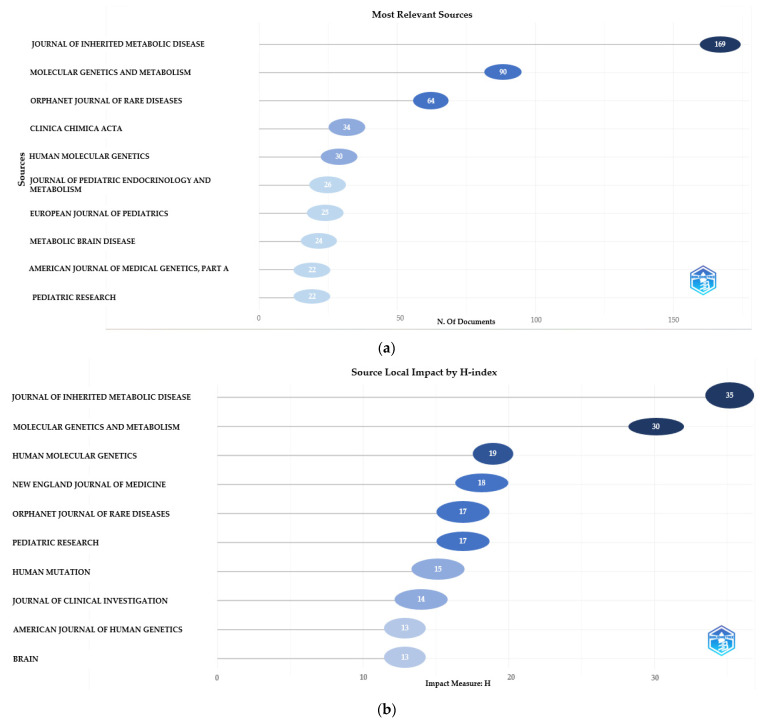
(**a**) The most relevant journals number of documents is shown. (**b**) Journals’ impact by H-index is shown.

**Figure 4 children-10-01205-f004:**
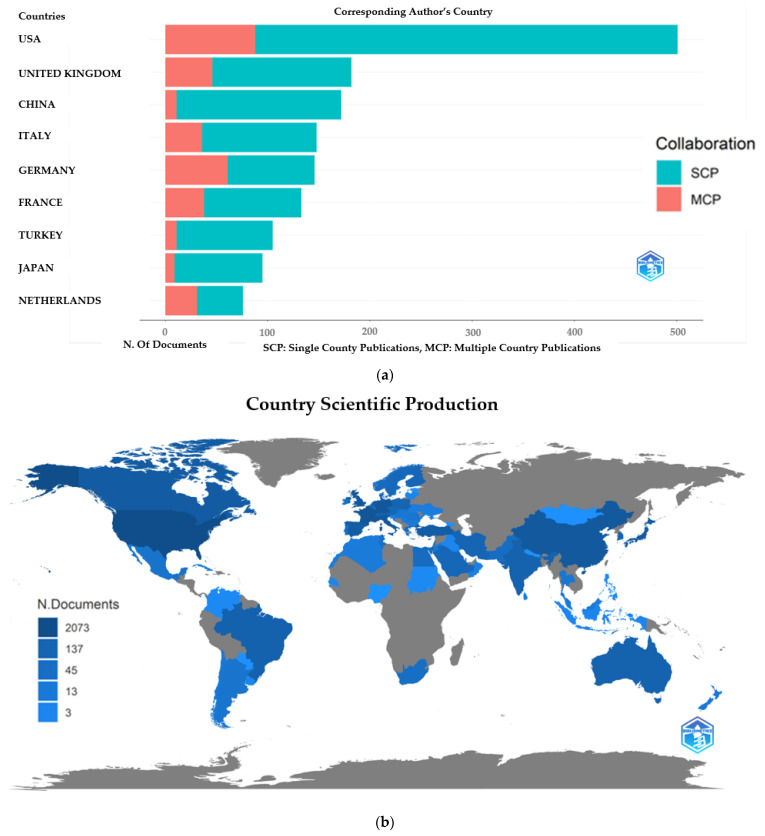
(**a**) The number of documents in the corresponding authors’ country and also single-country or multiple country publications are shown. (**b**) The distribution of countries by article productivity is shown. (**c**) The document weight of the countries of the co-authors by years and their relationship with each other are shown.

**Figure 5 children-10-01205-f005:**
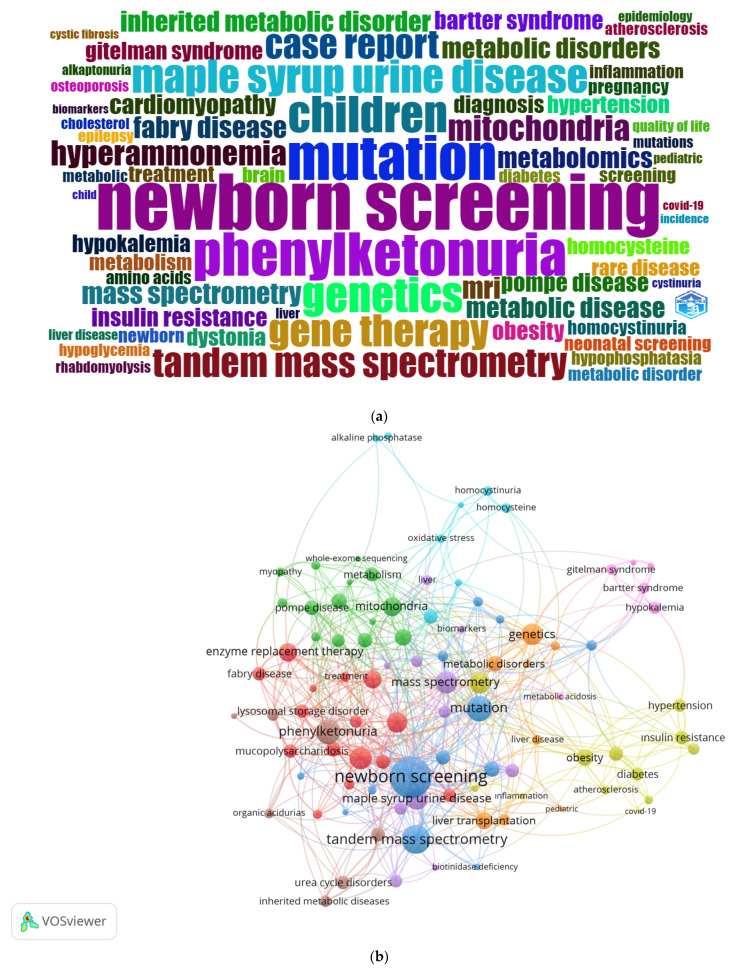
(**a**) The 100 most frequent keywords used by authors are shown. The most frequent keywords are shown bigger, and the less frequent keywords are shown smaller. (**b**) Co-occurrences of keywords are shown.

**Figure 6 children-10-01205-f006:**
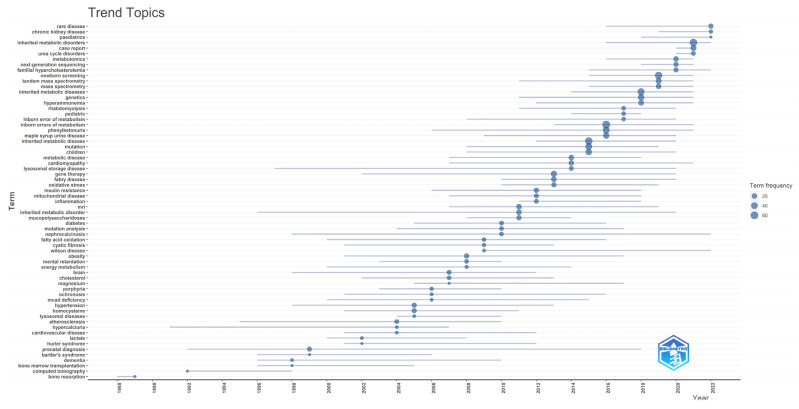
The most frequent keywords used by authors according to the years are shown.

**Table 1 children-10-01205-t001:** General information about the study.

Description	Results
Main Information about Data	
Timespan	1968:2023
Sources (Journals, Books, etc.)	1012
Documents	2702
Annual Growth Rate %	3.69
Document Average Age	14.2
Average citations per doc	26.4
References	88,136
Document Contents	
Keywords Plus (ID)	16,542
Author’s Keywords (DE)	5380
Authors Collaboration	
Authors	14,782
Single-authored docs	207
Co-Authors per Doc	6.97
International co-authorships %	21.06
Document Types	
Research article	2702

**Table 2 children-10-01205-t002:** Top 15 most commonly used keywords in IMD publications.

Keywords	Number of Articles
newborn screening	54
Mutation	43
Phenylketonuria	42
Children	35
Genetics	34
maple syrup urine disease	32
gene therapy	31
case report	28
tandem mass spectrometry	28
glycogen storage disease	27
Mucopolysaccharidosis	27
liver transplantation	25
Mitochondria	25
Hyperammonemia	24
enzyme replacement therapy	21

## Data Availability

The data presented in this study are available upon request from the corresponding author.

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
