# Peer review of "Inherited Metabolic Diseases from Past to Present: A Bibliometric Analysis (1968–2023)"

_children, 2023, doi:10.3390/children10071205_

Round 1

Reviewer 1 Report

The bibliometric analysis by Banu Kadıoğlu Yılmaz et al. was interesting. The current content was basically satisfactory. However, the reviewer strongly suggested to add some analyses using VOSviewer software. For instance:

· Co-authorship (nation, organization and authors)

· Bibliographic coupling

· Co-citation

· Co-occurrence (keywords and terms).

After that, please supplement some useful discussions per these analyses. The reviewer believed that the paper would be significantly improved then.

Besides, a minor suggestion was to translate the first Reference into English.

To sum up, a Minor Revision was needed.

Author Response

Answer to the first reviewer:

By using the VOSviewer software;

  • We analyzed and added “Figure 2d. Co-citations analysis of authors” and wrote our results under the title of “3.1.2 Distribution of Main Authors” in the “Results” section, which is shown in red color.
  • We analyzed and added “Figure 2e. Bibliographic coupling analysis of authors by years” and wrote our results under the title of “3.1.2 Distribution of Main Authors” in the “Results” section, which is shown in red color.
  • We analyzed and added “Figure 4c. The document weight of the countries of the co-authors by years and their relationship with each other” and wrote our results under the title of “3.1.4 Analysis of Corresponding Authors’ Countries” in the “Results” section, which is shown in red color.
  • We analyzed and added “Figure 5b. Co-occurrences of Keywords” and wrote our results under the title of “3.1.5 Analysis of Keywords” in the “Results” section, which is shown in red color.
  • We discussed these results in the “Discussion” section in paragraph 2 and paragraph 5, as shown in red.
  • We translated the first reference to the English version.

Reviewer 2 Report

This manuscript provides a bibliometric overview by conducting of a literature search on the keywords "inherited metabolic disease".

As the authors correctly describe, there are some limitations, especially the fact that no further keywords were included in the search such as "inborn errors of metabolism", one of the most frequently used terms in this field.

Further issues:

1. The use of English needs to be checked, especially in the Introduction.

2. The text and numbers in most figures (1, 2a, 2b, 2c, 3a, 3b, 4a, 4b, 6) are illegible, the font size should be markedly increased.

3. Fig. 2a+2b+2c: is the name "Wagner M" instead of "Wajner M"?

4. Discussion, Paragraph 5: this is rather speculative, please rephrase. Example: better like this: "We see the journey of the diagnostic approach in inherited metabolic diseases in which the detection of the accumulated substance is being complemented by enzymatic evaluation, molecular genetic studies, and metabolomics." (The latter three can be important but are not expected to replace detection of the accumulated metabolites in the foreseeable future.)

5. Conclusions: "With the development of molecular studies, we predict that treatment will change from enzymatic therapy to gene therapy in the future." This is speculative and not true for many disorders. Rephrase as "Gene therapy is expected to become more important in the future, complementing conventional therapies, diet and enzymatic therapy."

The use of English needs to be checked, especially in the Introduction.

Author Response

Answer to the second reviewer:

  • We used the keyword “inherited metabolic disease” for this bibliometric analysis. An important limitation is that we did not use the second keyword like, “inborn errors of metabolism.” We wrote this situation under discussion in the “Limitation” section.
  • We corrected the grammar mistakes in the article.
  • We increased the font size in figures 1, 2a,2b,2c,3a,4a,4b and 6.
  • In Figures 2a,2b, and 2c, the name of the author Wajner M, which is “Moacir Wajner” was true, but it was written by mistake as Wagner M in the results section in the “3.1.2. Distribution of Main Authors” part in line 4. We corrected the name of the author as Wajner M.
  • We rephrased the sentence in the discussion section, in paragraph 6, as “We see the journey of the diagnostic approach in inherited metabolic diseases in which the detection of the accumulated substance is being complemented by enzymatic evaluation, molecular genetic studies, and metabolomics.” which is shown in red color.
  • We rephrased the sentence in the conclusion section, as "Gene therapy is expected to become more important in the future, complementing conventional therapies, diet and enzymatic therapy." which is shown red color.